# Structural, Optical, Electric and Magnetic Characteristics of (In_1−x_Gd_x_)_2_O_3_ Films for Optoelectronics

**DOI:** 10.3390/ma16062226

**Published:** 2023-03-10

**Authors:** Moustafa Ahmed, Yas M. Al-Hadeethi, Ali M. Abdel-Daiem, Essam R. Shaaban

**Affiliations:** 1Department of Physics, Faculty of Science, King Abdulaziz University, Jeddah 21589, Saudi Arabia; 2Physics Department, Faculty of Science, Al-Azhar University, Assiut P.O. Box 71452, Egypt

**Keywords:** (In_1−x_Gd_x_)_2_O_3_ thin films, XRD, spectroscopic ellipsometer, optical parameters, electrical parameters, magnetic properties

## Abstract

After (In_1−x_Gdx)_2_O_3_ powder with a wide x range of 0 to 10 at.% was chemically produced, (In_1−x_Gdx)_2_O_3_ thin films were evaporated under ultra-vacuum using an electron beam apparatus. We investigated the influence of the Gd doping concentration on the magnetic, optical, electrical, and structural properties of the resultant In_2_O_3_ deposits. The produced Gd-doped In_2_O_3_ films have a cubic In_2_O_3_ structure without a secondary phase, as shown by the X-ray diffraction results. Additionally, the chemical analysis revealed that the films are nearly stoichiometric. A three-layer model reproduced the spectroscopic ellipsometer readings to determine the optical parameters and energy gap. The $E_{g}^{opt}$ changed toward the lower wavelength with growing the Gd doping in (In_1−x_Gdx)_2_O_3_ films. The $E_{g}^{opt}$ in the (In_1−x_Gd_x_)_2_O_3_ films was observed to increase from 3.22 to 3.45 eV when the Gd concentration climbed. Both carrier concentration and hall mobility were found during the Hall effect studies. It was possible to construct the heterojunction of Ni (Al)/n-(In_1−x_Gd_x_)_2_O_3_/p-Si/Al. At voltages between −2 and 2 volts, investigations into the dark (cutting-edge-voltage) characteristics of the produced heterojunctions were made. The oxygen vacancies and cationic defects in the lattice caused by the uncompensated cationic charges resulted in significant magnetism and ferromagnetic behavior in the undoped In_2_O_3_ films. The (In_1−x_Gd_x_)_2_O_3_ films, however, displayed faint ferromagnetism. The ferromagnetism seen in the (In_1−x_Gd_x_)_2_O_3_ films was caused by oxygen vacancies formed during the vacuum film production process. Metal cations created ferromagnetic exchange interactions by snatching free electrons in oxygen.

## 1. Introduction

Early advancements in diluted magnetic semiconducting (DMS) materials [1,2,3] were mostly based on II and IV semiconductors, such as CdSe and ZeSe, where the positive ion’s valence was 3+ and magnetic ions with this valence, such as Mn, were doped. However, the difficulty in converting these materials into positive- and negative-type semiconductors makes them less desirable for use. GaAs, a recently utilized III-V semiconductor, have been doped with the appropriate magnetic ions (Mn) impurity to enhance DMSs [4]. At ambient temperatures, these DMSs have no magnetic properties and low Curie temperatures (200 K). They are therefore inappropriate for applications that require room temperature. Fortunately, oxide semiconductors may help to bridge this gap. Recent research has demonstrated that ferromagnetism is seen in semiconductors with magnetic ion doping, such as zinc oxide [5], cerium oxide [6], tin oxide [7], and indium oxide [8], even at room temperature. Due to its broadband gap and n-type semiconductor characteristics, indium oxide (In_2_O_3_) is a prime candidate among these oxides of semiconductor. Optoelectronic industries make heavy use of In_2_O_3_ [9]. The ferromagnetism that conducts In_2_O_3_ might add more bristles to the hat. Doping the In_2_O_3_ matrix lattice with a variety of magnetic ions, such as Cr [10], Mn [11], Co [12], Fe [13], and Ni [14], allowed the creation of In_2_O_3_ ferromagnetic materials. This claim describes the deposition and characterization of translucent and conducting In_2_O_3_ films that have been gadolinium oxide doped with Gd. In order to create DMS oxides, we investigate in this study if it is feasible to dope Gd into the In_2_O_3_ host lattice. Here, we create thin film samples and powdered Gd-doped In_2_O_3_ (dilute concentration). Systematic analysis of materials employs measurements of their structure, morphology, electrical characteristics, magnetism, and spectroscopic ellipsometry (SE). On the other hand, for voltages extending from −2 to 2 volts, the (J-V) properties of the Ni (Al)/n-(In_1−x_Gd_x_)_2_O_3_ /p-Si/Al heterojunction are investigated.

## 2. Details of the Experiment

The polycrystalline composites (In_1−x_Gd_x_)_2_O_3_ with x = 0, 2, 4, 6, 8, and 10 at.% were created by combining analytical powders of Gd_2_O_3_ and In_2_O_3_ (Sigma-Aldrich purity 99.998%) using the following equation:(Gd_2_O_3_) + (1 − x) In_2_O_3_ → (In_1−x_Gd_x_)_2_O_3_

The granules were chaotically mixed in a robotic pestle and mortar at RT for 45 min. This mixture was subsequently compressed into disc-shaped grains that were used as the base for films. A clean Corning #1022 glass substrate was covered in (In_1−x_Gd_x_)_2_O_3_ thin films using an electron beam gun evaporation (Edward 306Auto) deposition system in a high vacuum environment (see Figure 1). Before evaporation, we fill the graphite boat with (In_1−x_Gd_x_)_2_O_3_ powder, and a typical vacuum of about comparable to 9 × 10^−7^ mbar was applied. The substrate temperature was maintained at 140 °C while the deposition rate was held at 2 nm/min in order to increase film adhesion. With the use of Cu-K1 radiation from a Philips X-ray diffractometer (type X’pert), the phase of (In_1−x_Gd_x_)_2_O_3_ films was estimated. Pure silicon with a purity of 99.9999 was used to calibrate the XRD apparatus. The fundamental makeup of the films was examined using an X-ray spectrometer (EDXS) and a SEM (JOEL XL) at a voltage of 30 kV. Using a compensator (J-A-Woollam, M-2000, QDI: Darmstadt, Germany) in the wavelength range of 300 to 1100 nm, the ellipsometry parameters of the (In_1−x_Gd_x_)_2_O_3_ layer were measured. A 70° angle was used to collect data on ellipsometry. Using the WVASE32-program from J-A-Woollam Corporation, detailed modeling was performed to determine the optical constants of the (In_1−x_Gd_x_)_2_O_3_ films. The transmission of the films was evaluated using a JASCO-670 UV-Vis-NIR spectrophotometer. The Van der Pauw method and a Hall effect measurement method were utilized to evaluate the electrical properties of (In_1−x_Gd_x_)_2_O_3_ films (HMS-5000, ECOPIA, Gyeonggi, Korea). On a 1 cm^2^ glass substrate, resistivity, mobility, carrier type, and carrier concentration were all assessed. The produced device’s current density–voltage (J-V) was conventionally measured using Keithley 610 and 617 voltage source and current meters to determine the current density across the heterojunction at various Gd contents. The dark (current density–voltage) characteristics were measured at room temperature in a fully dark environment.

Finally, a vibrating sample magnetometer model was used to examine the films’ magnetic characteristics (VSM model-9600M-1, Lowell, MA, USA). At RT, magnetic measurements were made.

## 3. Results and Discussion

### 3.1. X-ray Diffraction and Morphology

The measured XRD pattern of the In_2_O_3_ powder is shown in Figure 2a as described by the strength of the peak at a particular angle. The picture displays the peak positions that the X’Pert-HighSore program was able to collect following code 06-0416. Figure 2b shows the XRD patterns of the (In_1−x_Gd_x_)_2_O_3_ films with x = 0, 2, 4, 6, 8, and 10 at.%.

According to Figure 2b, the strength of the peak at a specific angle best describes the measured XRD pattern of the In_2_O_3_ powder. This figure shows the peaks that the X’Pert-HighSore program was able to record according to code 06-0416. The XRD patterns of the (In_1−x_Gd_x_)_2_O_3_ films with x = 0, 2, 4, 6, 8, and 10 at.% are shown in Figure 2b. All films have a clear focus (222). JCPDS card No. 00-006-0416 states that the characteristic diffraction peak of an In_2_O_3_ film at 2θ = 30.57, which has a preferred placement in the (222) plane, confirms the existence of a cubic structure. The films’ XRD did not reveal the presence of Gd_2_O_3_. The XRD peaks gradually decrease as more Gd_2_O_3_ is added to the film, which is brought on by a decrease in crystallinity.

The XRD peaks with the indication (222) are transferred to the higher diffraction angle and are magnified in Figure 2c due to the higher ionic radii of Gd (1.05) than of In (0.94). In accordance with [15,16], the lattice constant “*a*”, the plane’s characteristic (*hkl*), and the interplanar spacing (*d_hkl_*) have the following relationships:(1)$$d_{hkl}=\frac{a}{\sqrt{h^{2}+k^{2}+l^{2}}}$$

Using Bragg’s law, the spacing *d_hkl_* is related to the Bragg’s diffraction angle *θ* as
(2)$$\lambda =2d_{hkl}\sin\theta$$

As the concentration of Gd increases, the estimated lattice parameters *a* = *b* = *c* shrink and are in good agreement with the JCPDS data. The decrease in lattice parameters could be attributed to the variation in the ionic radii of Gd and In, which causes a lattice deformation.

#### Crystal Size and Lattice Strain

For (In_1−x_Gd_x_)_2_O_3_ films, the average crystallite size *D* is calculated using Scherer’s equation [17,18]
(3)$$D_{v}=\frac{k\lambda }{\beta \cos(\theta )}\;$$

The average crystallite size of (In_1−x_Gd_x_)_2_O_3_ films dropped as the concentration of Gd dopant increased because the rise of Gd ions caused the In_2_O_3_ matrix to deform. The nucleation and growth rates of Gd-doped In_2_O_3_ films may be constrained as a result. The calculated values for crystallite size are shown in Table 1. The following equation is used to determine the dislocation density (δ) of the films based on the experiment’s findings [19]:δ = 1/*D*^2^(4)

The worth of the film and its defect structure are determined by the dislocation density of the films. The In_2_O_3_ matrix is given Gd doping in this study, and as a result, the density of dislocations rapidly increases, exposing the structure of the imperfections.

The Stoke and Wilson equation [19,20] can be used to compute the lattice strain (ε).
(5)$$\varepsilon =\frac{\beta }{4\tan\theta }$$

*β* can be corrected by the subsequent association.
(6)$$\beta =\sqrt{\beta_{obs}^{2}-\beta_{std}^{2}}$$
where $\beta_{obs}$ is the peak width of the film, and $\beta_{std}$ is the standard peak width (single crystal silicon). The films of (In_1−x_Gd_x_)_2_O_3_ with (x = 0, 2, 4, 6, 8, and 10 at.%), as well as the crystallite size and lattice strain, are shown in Figure 3. The lattice strain increases as the Gd inclusion increases and the crystallite size decreases. The ionic radius of Gd is considerably higher than that of In, which leads in a reduction in crystalline size and an increase in lattice strain.

The observed decrease in *D* and increase in Gd may also be attributed to two additional factors: first, the combined effect of the host In_2_O_3_ crystal’s lattice distortion brought on by the substitution of higher ion-sized Gd atoms [21]; and second, the surface of the doped samples was claimed to have developed a thin Gd-O-In layer as a result of too many In ions in the precipitation solution inhibiting crystal formation [21,22].

Figure 4a,b displays the EDAX spectrum of thin films of (In_0.98_Gd_0.02_)_2_O_3_ and (In_0.90_Gd_0.10_)_2_O_3_ with weight percent and atomic percent. Strong In and O signals are visible in the EDAX spectrum, however (In_0.98_Gd_0.02_)_2_O_3_ clearly shows modest Gd signals due to the rising Gd concentration. Additionally, the EDXS investigation’s findings demonstrated that the composition is nearly stoichiometric.

SEM pictures of the In_2_O_3_ and (In_0.90_Gd_0.10_)_2_O_3_ thin films created on glass substrates are shown in Figure 5a,b. The pictures show both films’ high adhesion and thick, erratic structure. In terms of the histograms (in Figure 6c,d of Figure 5a,b, the average grain size distribution was calculated using Gauss curve fitting and was discovered to be 40 nm and 25 nm for x = 0.0 and 0.1, respectively, as shown in the histogram of Figure 5a,b. These numbers are higher than the XRD-measured crystallite size. The XRD measurement revealed coherent X-ray diffraction at the crystal areas. However, the SEM examination’s grain size was established between the grain boundaries. AFM enables the acquisition of a quantitative validation of the film’s structural issues. Standard 3D dimension AFM scan pictures are displayed in Figure 6a–c for the In_2_O_3_, (In_0.94_Gd_0.06_), and (In_0.90_Gd_0.10_)_2_O_3_ thin films, respectively. The small shapes appearing in Figure 6 may be attributed to the presence of the doping Gd in addition to the matrix material In_2_O_3_.

The RMS is calculated as the root mean square of the measured surfaces of the microscopic peaks and valleys, i.e., it represents the root mean square value of ordinate values within the defined area. It is comparable to the height of the standard deviation. The peaks and valleys of the rooftops are measured individually for each value, but the measurements are applied to a separate formula. Examining the calculations reveals that the RMS value is impacted by a single significant peak or fault inside the microscopic surface texture. In reference [22], more information regarding the trend of the RMS values derived by the AFM image is provided.

The XRD study size and the grain size estimated by AFM follow a similar pattern, however, the computed value is greater than the predicted crystal size. These root mean square (RMS) values for film roughness were obtained from the evaluation: 5.47, 5.41, and 5.32 nm.

### 3.2. Spectroscopic Ellipsometry for Measuring Optical Parameters

The refractive index (*n*) and absorption index (*k*) of the (In_1−x_Gd_x_)_2_O_3_ with (x = 0, 2, 4, 6, 8, 10 at.%) films have been extracted by spectroscopic ellipsometry SE. The relationship between these films’ microstructure and optical characteristics and the SE gaining parameters *ψ* and Δ and is shown in the table below [23,24].
(7)$$\rho =\frac{r_{p}}{r_{s}}=\tan\psi \exp\left(i\Delta \right)$$
where *r*_s_ and *r*_p_ stand for the relative Fresnel coefficients of reflection from the film layer.

The measurements of *ψ* and Δ on the (In_1−x_Gd_x_)_2_O_3_ films at a 70° incidence angle are shown in Figure 7a,b. In order to calculate the films’ *n*, *k*, and film thickness *d*, a three-layer optical model was utilized. The model consists of three layers: the substrate, the “B-spline” (In_1−x_Gd_x_)_2_O_3_ layer, and the rough layer. Changes in *ψ* and Δ were fit using the mean root square (MRS) function and least square regression as follows:(8)$$\sigma =\frac{1}{2N-M}\sum_{i=1}^{N}\left({\left(\frac{\psi_{i}^{\mathrm{mod}}-\psi_{i}^{\exp}}{\sigma_{\psi ,i}^{\exp}}\right)}^{2}+{\left(\frac{\Delta_{i}^{\mathrm{mod}}-\Delta_{i}^{\exp}}{\sigma_{\Delta ,i}^{\exp}}\right)}^{2}\right)$$

Figure 8a,b depicts the modeled spectrum dependency of *ψ* and Δ for (In_0.90_Gd_0.10_)_2_O_3_ films and it shows a superior fit with the measured data (symbols) acquired across the whole range. The average thickness for all films is 200 ± 1.23 nm, while the average surface roughness is 5.67 ± 0.13 nm. Surface roughness obtained through SE can match that of a standard 3D dimensional AFM scan image in terms of quality. More information on SE execution and its conceptual B-spline optical model is available in reference [23].

In Figure 9a,b, respectively, the computed values of *n* and *k* for the (In_1−x_Gd_x_)_2_O_3_ films are displayed. These two graphs demonstrate how the n and k rapidly decrease as the concentration of Gd_2_O_3_ rises. The (In_1−x_Gd_x_)_2_O_3_ layer’s absorption coefficient and absorption index are connected by the equation (k = αλ/4π).

The peak width of the refractive index (700 < λ < 800) in Figure 9a is determined by the transmission of light via nano-holes and the negative phase shift brought on by surface plasmons (SP) scattering at the interfaces between the nano-hole and the substrate. SPs, which are collective oscillations of metal-free electrons trapped at metal–dielectric interfaces and stimulated by an electromagnetic field that is incident on them, are confined in the metal surface. The multiple optical resonance peaks that appear as a shoulder at 750 nm are caused by the coupling and decoupling process between the SP resonance evanescent waves and the incident light through the nano-hole. Reference [24] provides additional information on SPs and plasmon resonance. The refractive index and extinction coefficient for all samples decreases as the wavelength increases, as seen in Figure 9a,b. Light scattering and the decline in absorbance are the causes of this phenomenon. The refractive index and extinction coefficient in the visible region decreases as the Gd content increases. The extinction coefficient value in Figure 9b is rather high. This demonstrates the substantial dielectric loss of the Gd-doped indium oxide thin films. The polycrystallanity of the films was indicated by ripples (interference patterns) in the extinction coefficient spectrum in the wavelength range of 400 nm to 800 nm. The nano-hole form and nano-hole periodicity allow for exact control of the transmission wavelength location and intensity. For instance, the contribution from the structural margins becomes increasingly substantial in short-range systems with few holes, resulting in unique emission patterns.

The optical transitions of the materials being studied in the high absorption zone (α ≥ 10^4^) are provided by the Tauc equation illustrated below [25,26].
(9)$$\alpha .h\nu =\frac{K^{\prime }{\left(h\nu -E_{g}^{opt}\right)}^{x}}{h\nu }$$
where $K^{\prime }$ is known as the Tauc parameter which represents the degree of disorder in the materials and depends on the transition probability, *x* is the super index controlled by the transition type that controls optical absorption, and $E_{g}^{opt}$ is the energy gap.

When we schemed (α*hν*)*^x^* vs. (*hν*) for (In_1−x_Gd_x_)_2_O_3_ films with different Gd contents, a direct transition was confirmed. The outcomes are shown in Figure 9c, which represents the intersection of the linear component’s extended linear fit and the energy axis. As can be observed in Figure 10, the findings demonstrate that the obtained $E_{g}^{opt}$ values rise as the Gd concentration does.

As the amount of Gd_2_O_3_ rises as a result of the continual substitution of In atoms with Gd, the fundamental band gap moves to the blue. This increase in the carrier electron injection results in the Burstein–Moss effect, which shifts the Fermi level and combines into covalent bands [27]. The ZnO crystal develops more interstitial oxygen impurities when the doping concentration is increased, which results in more delocalized states, a smaller band gap, and a lower energy gap.

The measured transmittance (*T*) of films formed of (In_1−x_Gd_x_)_2_O_3_ covering the 300 to 1100 nm spectral region is shown in Figure 11. High-transparency films are needed for optical devices as the transmittance increases in comparison to In_2_O_3_ films as the Gd doping level rises [27,28]. The more oxygen distributed across the coating is what causes the increase in transmittance [29,30,31]. The (*T*) trend of the (In_1−x_Gd_x_)_2_O_3_ films in the strongly absorbing area, where the transmittance changes to blue with increasing Gd concentration, is shown in the inset of Figure 11. As a result, the energy gap expands, as seen in Figure 11.

### 3.3. Electric Properties

Using a standard four-point probing technique, the electrical characteristics of the (In_1−x_Gd_x_)_2_O_3_ layer films with different Gd contents were investigated. *R*_s_ = 4.53.V/I [Ω/sq], where V is the voltage in volts, I is the current in amperes, and the number 4.53 is the correction constant, is the equation used to calculate sheet resistance [32,33]. In accordance with the equation ρ = *R*_s_ *d*, the relationship between resistivity (measured in ohm cm) and *R*_s_ is accurate if the film thickness is *d* [34]. As the Gd content increases, the resistivity decreases, as shown in Figure 12. The measurements show that the (In_1−x_Gd_x_)_2_O_3_ films are n-type materials. The figure displays the measured electrical properties of the prepared (In_1−x_Gd_x_)_2_O_3_ thin films as a function of Gd concentration. It is obvious that the concentration, n, and mobility of the carriers alter as the Gd content grows. For both carrier concentrations, the mobility is optimum at a Gd concentration of 8 at.% and becomes approximately fixed at 10 at.%. These results demonstrate that resistance decreases as carrier concentration rises. An increase in mobility is associated with a decrease in crystalline size and an increase in lattice strain.

#### Current Density versus Voltage for Ni (Al)/n_−_(In_1−x_Gd_x_)_2_O_3_/p-Si/Al Heterojunction

Figure 13 displays the analyzed p-n junction diagram. It is important to note that the main elements affecting how the created p-n junction responds to reverse and forward bias employed in the vicinity of (−2 to 2 volts). The applied voltage affects the current density *J*, and the subsequent equation [35,36] provides the other diode production specifications:(10)$$J=J_{0}\left(e^{\frac{qV}{n_{1}k_{B}T}}-1\right)$$

The electronic charge that equals $\left(1.6\times 10^{-19}\;C\right)$ and evaluates the Boltzmann’s constant $k_{B}$ in this equation at room temperature (RT) represents the saturation current density and typifies the quality factor of the manufactured diode. 

With applied voltage in the recommended ranges, Figure 14 shows the dark (J-V) characteristics of the produced diode in forward and reverse bias in (In_1−x_Gd_x_)_2_O_3_ thin films on silicon substrate. It is obvious that up until a level of 8%, the current density rises along with the Gd content before beginning to significantly be fixed at a level of 10%. Figure 14a,b depict the relationship between the forward and reverse biases of the applied voltage in the dark and in the light, respectively. The forward bias voltage’s current is higher than the reverse bias voltage’s current (J-V). These figures show how an increase in the forward bias behavior of the solar cell results in an increase in the current density behavior, which significantly increases in the low voltage area. In the depletion zone, also known as the “low voltage region”, the reverse current density of the examined produced p-n junction displays a weaker exponential behavior than the junction’s forward current density does in the same region. As a result, it may be argued that the constructed p-n junction has amazing rectification qualities because as the resistivity falls, the Gd concentration rises. The measurements demonstrate the n-type nature of the (In_1−x_Gd_x_)_2_O_3_ films. It is clear that when the Gd content rises, the carriers’ concentration, n, and mobility change. The mobility is greatest for both carrier concentrations at a Gd concentration of 8 at.% and becomes roughly fixed at 10 at.%. These findings show that the resistance diminishes with increasing the carrier concentration. A decrease in the crystalline size and an increase in the lattice strain are related to an increase in mobility. For instance, (In_0.92_Gd_0.08_)_2_O_3_ films that are candidates for optoelectronic and solar cell applications have fair crystal light size, high conductivity, high carrier concentration, and carrier mobility.

### 3.4. Magnetic Characterization

Figure 15 displays the magnetic hysteresis loops of In_2_O_3_ powder, which represent the diamagnetic behavior at room temperature (a). Figure 15b displays the magnetic loops of powder samples of (In_1−x_Gd_x_)_2_O_3_ with x = 2, 4, 6, 8, or 10 at.%. Small amounts of ferromagnetism are present in the samples of (In_1−x_Gd_x_)_2_O_3_ with (x = 2, 4, 6, 8, or 10 at.%), and the magnetization increases with the magnetic field. It implies the presence of paramagnetic phases in Gd doped In_2_O_3_ powders. Our results unmistakably showed that there were no signs of ferromagnetic clusters or impurity phases. When the In3+ lattice site of the Gd3+ is occupied, a single-phase structure is produced. The oxygen vacancy has therefore very little probability of growing.

Figure 16 displays the magnetization vs field-dependent observed at 300 K curves for the (In_1−x_Gd_x_)_2_O_3_ and (x = 0, 2, 4, 6, 8, 10 at.%) films. A revised removal of the substrate contribution yielded the data for the magnetization films. At room temperature, soft ferromagnetic characteristics were present in all doped films. It is important to note that In_2_O_3_ films, even when undoped, display ferromagnetic properties and unique magnetization. However, in undoped semiconducting oxide films such as TiO_2_, ZnO, HfO_2_, and In_2_O_3_, ferromagnetism is present. This may be because of oxygen vacancies and cation defects in the lattice, which may be the cause of the uncompensated cation charge. As vacancies, spin splitting, and high spin states develop in this system, ferromagnetism and the electrons filling the oxygen vacancies engage in exchange [37]. Density functional theory (DFT) calculations on magnetism demonstrate that intrinsic point defects such as “O” vacancies and In interstitials serve as shallow donors while, in contrast, “O” gaps and In vacancies act as shallow acceptors [38]. Therefore, uncompensated cations caused by defects such as oxygen vacancies and grain boundary defects can be responsible for the ferromagnetic hysteresis loops observed in undoped In_2_O_3_ films. Additionally, at ambient temperatures, all Gd-doped In_2_O_3_ films display ferromagnetism, and the saturation magnetization changes nominally as the Gd doping concentration increases. As a result, we explain the observed ferromagnetism as the result of the creation of magnetopolarons as a result of oxygen vacancies and trapped electrons in Gd-doped In_2_O_3_ films, which leads to room temperature ferromagnetism [39]. The oxygen vacancies produced during film deposition may be the cause of the observed room temperature ferromagnetism in films. replying to Coey and colleagues. According to [40], oxygen vacancies in wide-band gap semiconductors will trap free electrons; these trapped electrons (F centers) will then act as an intermediary in magnetic exchange interactions with nearby metal cations. Additionally, numerous theoretical and experimental studies demonstrated that cationic vacancies are the root cause of ferromagnetism [41,42].

With increasing Gd concentration, (In_1−x_Gd_x_)_2_O_3_ films become more ferromagnetic at (x = 2, 4, 6, 8, or 10 at.%). The magnetic exchange between Gd^+3^ and In^3+^ surrounding the empty electron trap may be the cause of the enhanced ferromagnetism. Ferromagnetism results from the magnetic exchange of Gd^3+^ pairs at the Gd^3+^-F core. As Gd content rises, the saturation magnetization rises as well.

## 4. Conclusions

Using a chemical reaction process, (In_1−x_Gd_x_)_2_O_3_ powder with x = 0, 2, 4, 6, 8, and 10 at.% was produced. After that, thin layers were evaporated in a high vacuum with an electron gun. The impact of the Gd doping level on the films’ structural, optical, and magnetic characteristics was examined. All of the Gd-doped In_2_O_3_ thin films showed the cubic In_2_O_3_ structure without any Gd-dopant impurity phases, according to the XRD data. Using an ellipsometric model, the optical constants of the (In_1−x_Gd_x_)_2_O_3_ films were determined from the SE measurements. With increasing Gd_2_O_3_ concentration, it was found that the entire spectrum range for both n and k decreased. This was attributed to the crystallinity contracting and growing lattice strain. With increasing Gd_2_O_3_ content, the energy gap widened from 3.22 eV to 3.45 eV, resulting in a direct optical transition. This was explained by the effects of the increased lattice strain and the reduction in grain size from 26.4 to 13.2 nm. The electrical resistivity of the (In_1−x_Gd_x_)_2_O_3_ films was demonstrated to decrease as more Gd was added. Conclusion: The (In_0.92_Gd_0.08_)_2_O_3_ film is one of the most promising materials for optoelectronic and solar cell applications since conductivity and transparency in the visible region both improve with increasing Gd concentrations. However, a thorough examination of the resulting forward and reverse biases has been conducted. When the bias is forward, the produced p-n junction or solar cell behaves differently, and this difference is most noticeable at low voltages. The growth and formation of the depletion area between the (In_1−x_Gd_x_)_2_O_3_ layer and the Si substrate are thought to be the cause of the exponential behavior at low voltage. It was noted that because of oxygen vacancies and cation defects in the matrix, the undoped In_2_O_3_ films display a ferromagnetic behavior with different magnetization. The Gd-doped In_2_O_3_ thin films, on the other hand, showed RTFM. As the Gd concentration in (In_1−x_Gd_x_)_2_O_3_ thin films increased, the ferromagnetic strength also increased. The ferromagnetic exchange between two neighboring Gd^3+^ via trapped oxygen vacancies was identified as the source of the observed ferromagnetism. As a result, Gd-doped In_2_O_3_ films are suitable candidates for spintronic device production at ambient temperatures.

## Figures and Tables

**Figure 1 materials-16-02226-f001:**
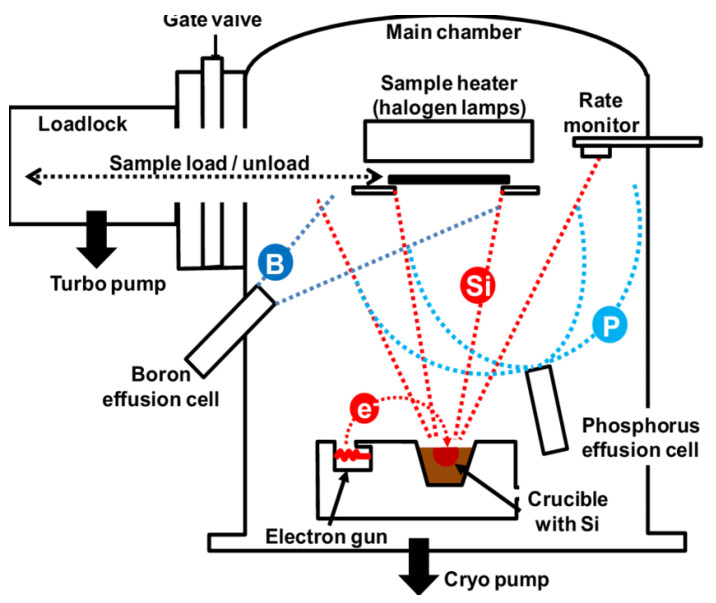
Scheme of the electron beam evaporation set.

**Figure 2 materials-16-02226-f002:**
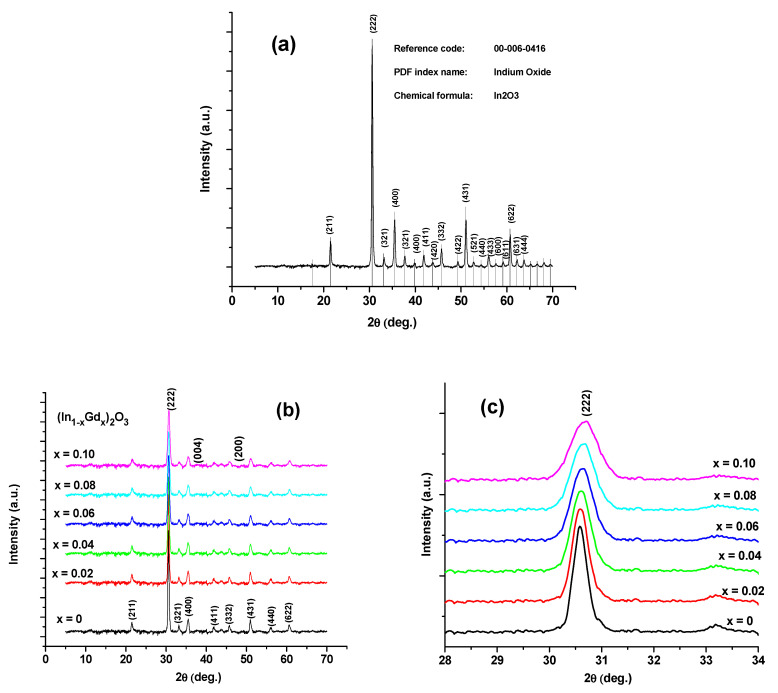
XRD of (**a**) In_2_O_3_ powder, (**b**) (In_1−x_Gd_x_)_2_O_3_ thin films, and (**c**) amplification of XRD peaks with index (222).

**Figure 3 materials-16-02226-f003:**
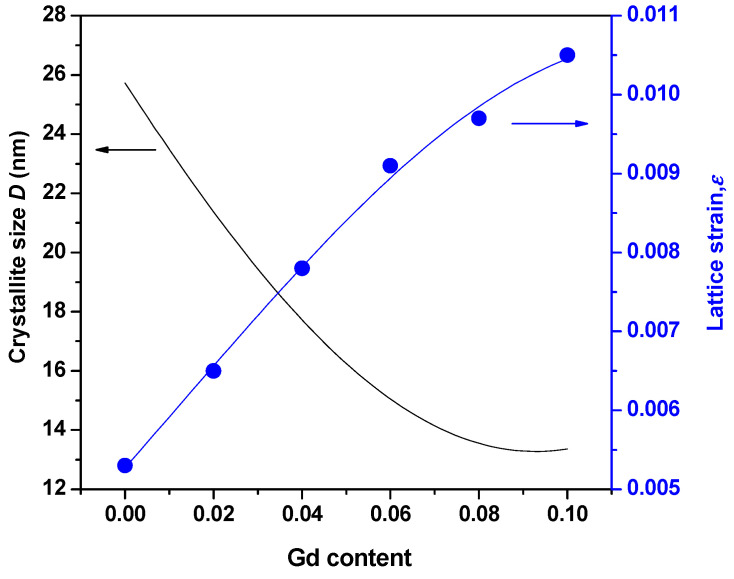
Variations in the lattice strain and crystalline size of (In_1−x_Gd_x_)_2_O_3_ films with the Gd content.

**Figure 4 materials-16-02226-f004:**
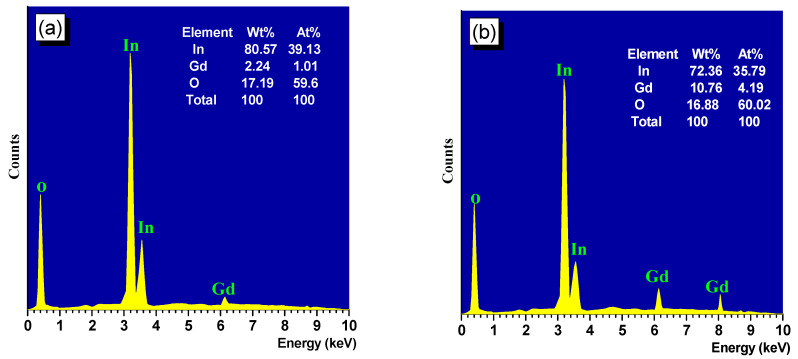
EDAX of (**a**) (In_0.98_Gd_0.02_)_2_O_3_ and (**b**) (In_0.90_Gd_0.10_)_2_O_3_ thin films.

**Figure 5 materials-16-02226-f005:**
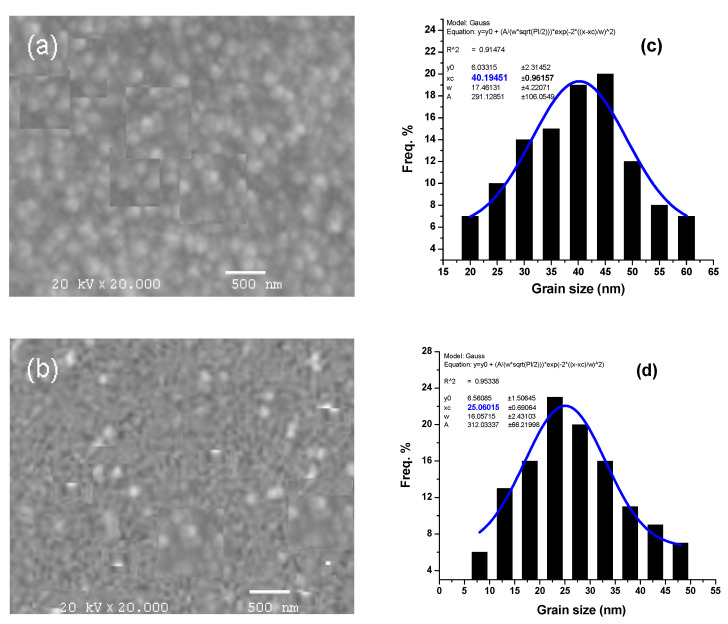
SEM of (**a**) In_2_O_3_ and (**b**) (In_0.90_Gd_0.10_)_2_O_3_ thin films and (**c**,**d**) their histograms.

**Figure 6 materials-16-02226-f006:**
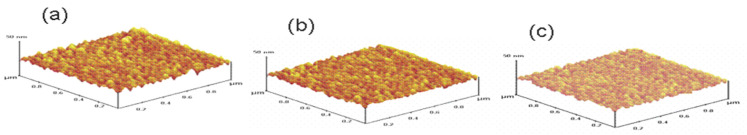
AFM of (**a**) In_2_O_3_ and (**b**) (In_0.94_Gd_0.06_)_2_O_3_ thin films and (**c**) (In_0.90_Gd_0.10_)_2_O_3_ thin films.

**Figure 7 materials-16-02226-f007:**
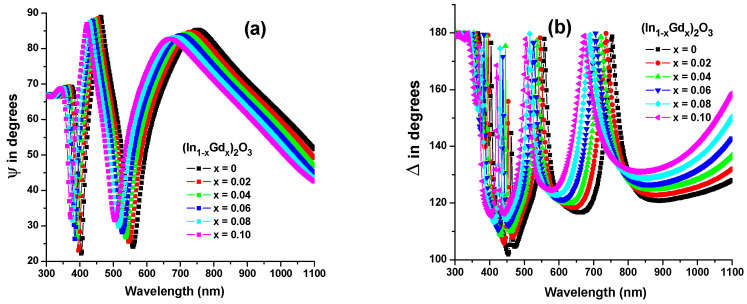
Ellipsometric parameters (**a**) *ψ* and (**b**) Δ versus wavelength for (In_1−x_Gd_x_)_2_O_3_ thin films.

**Figure 8 materials-16-02226-f008:**
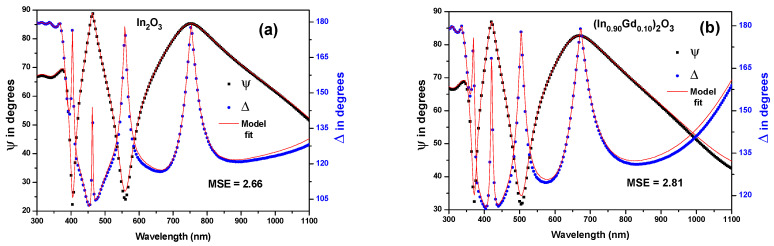
Ellipsometric spectra (*ψ* and Δ) of: (**a**) In_2_O_3_, and (**b**) (In_0.90_Gd_0.10_)_2_O_3_ films. The symbols represent the experimental measurements, and the solid lines plot the fitting model.

**Figure 9 materials-16-02226-f009:**
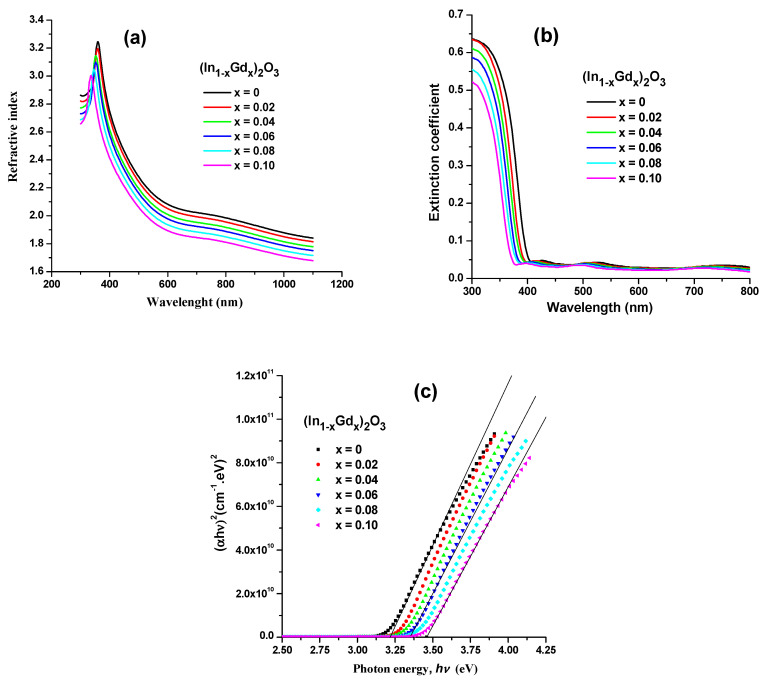
Spectra of (**a**) refractive index (**b**) the extinction coefficient for (In_1−x_Gd_x_)_2_O_3_ thin films and (**c**) plot of (*αhν*)^2^ versus *hν* for (In_1−x_Gd_x_)_2_O_3_ thin films.

**Figure 10 materials-16-02226-f010:**
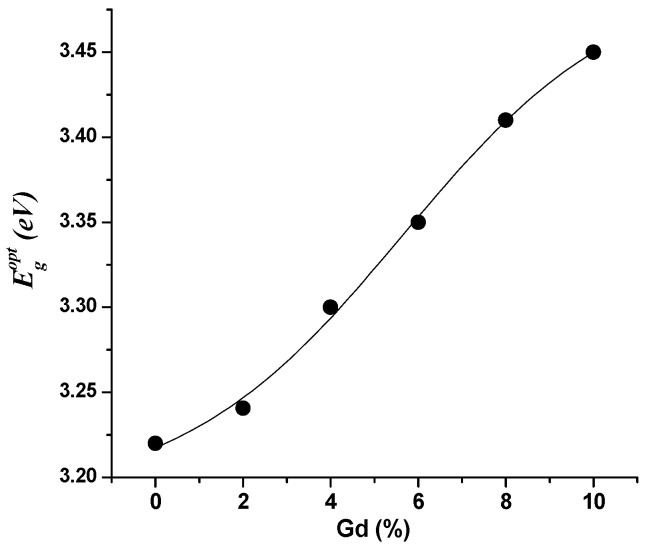
Plot of the optical energy gap versus the Gd content.

**Figure 11 materials-16-02226-f011:**
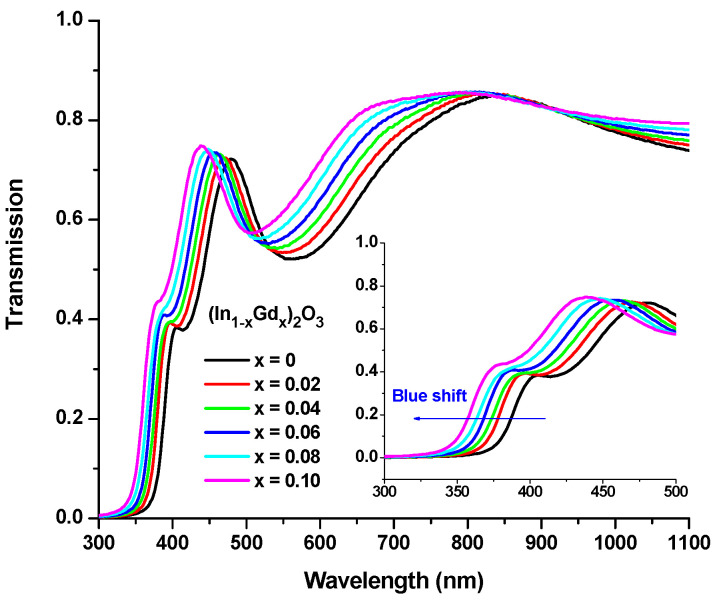
Spectra of transmission and reflection of (In_1−x_Gd_x_)_2_O_3_ films, the inset corresponds to the strong absorption region of transmission.

**Figure 12 materials-16-02226-f012:**
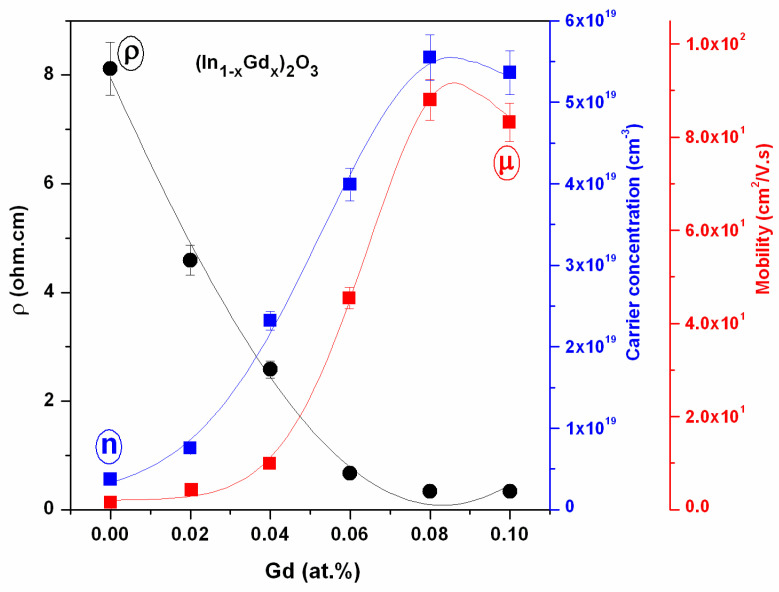
Electrical parameters in terms of Hall measurements as a function of Gd content for (In_1−x_Gd_x_)_2_O_3_ thin films.

**Figure 13 materials-16-02226-f013:**
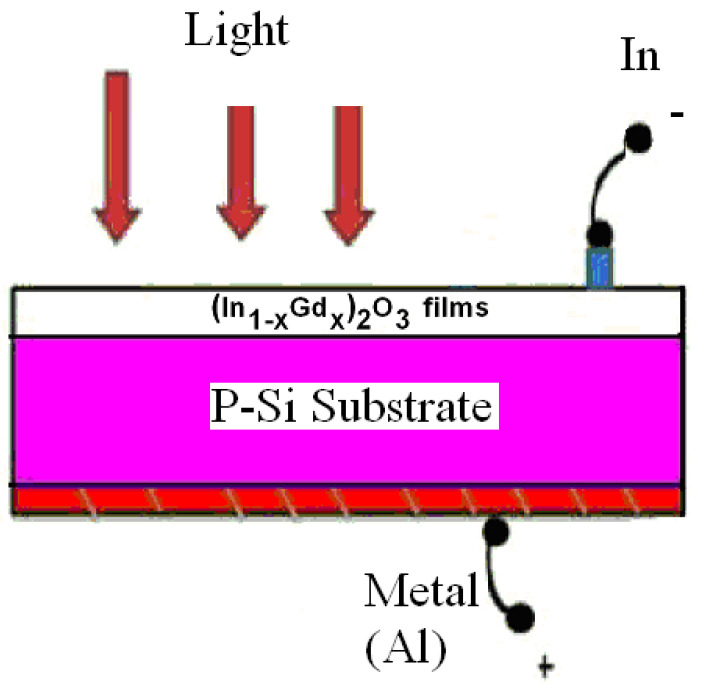
The diagram of studied p-n junction n- (In_1−x_Gd_x_)_2_O_3_ films/p-Si substrate.

**Figure 14 materials-16-02226-f014:**
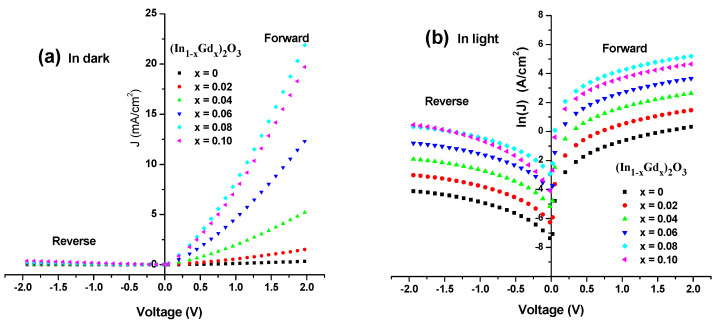
Current density, J versus voltage, V (**a**) in dark and (**b**) in light for n-(In_1−x_Gd_x_)_2_O_3_ films/p-Si substrate.

**Figure 15 materials-16-02226-f015:**
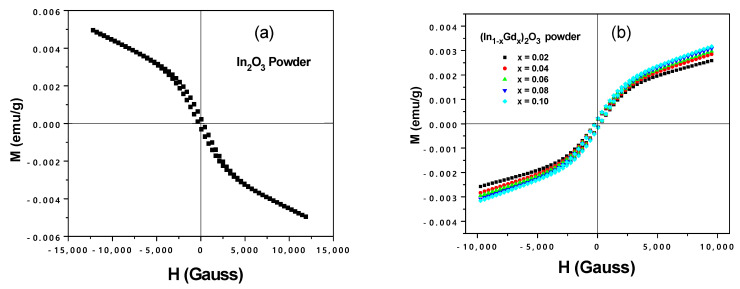
Magnetic hysteresis loops of (**a**) In_2_O_3_ and (**b**) (In_1−x_Gd_x_)_2_O_3_ (x = 0.02, 0.04, 0.06, 0.08, and 0.10) powders at room temperature.

**Figure 16 materials-16-02226-f016:**
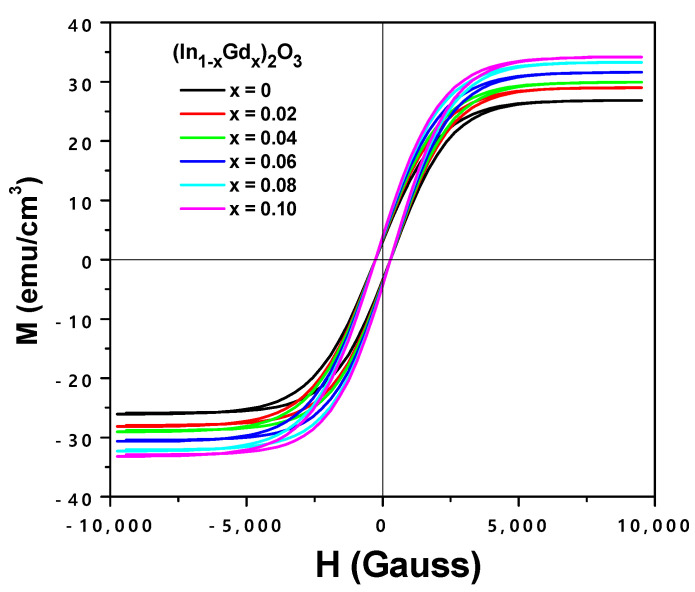
M-H curve for (In_1−x_Gd_x_)_2_O_3_ thin films.

**Table 1 materials-16-02226-t001:** Structural parameters for (In_1−x_Gd_x_)_2_O_3_ thin films.

Conc. x%	2θ	d-Space (Å)	Lattice Constant a (Å)	Corrected (FWAM) *β*	Crystallite Size *D* (nm)	Lattice Strain ε	Dislocation Density δ × 10^−4^ (nm)^−2^
0	30.5665	2.9212	10.1193	0.3301	26	0.0053	15
2	30.595	2.9185	10.1101	0.4111	21	0.0065	23
4	30.6116	2.917	10.1048	0.4902	18	0.0078	32
6	30.6523	2.9132	10.0917	0.5701	15	0.0091	44
8	30.6862	2.9101	10.0808	0.6112	14	0.0097	50
10	30.7121	2.9077	10.0725	0.6621	13	0.0105	59

## Data Availability

Not applicable.

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
