# Peer review of "Structural, Optical, Electric and Magnetic Characteristics of (In1−xGdx)2O3 Films for Optoelectronics"

_materials, 2023, doi:10.3390/ma16062226_

Round 1

Reviewer 1 Report

In this manuscript, the authors investigated the properties of Gd-doped indium oxide thin films with the different doping concentrations. When the concentration increases from 0 to 0.1, the grain size, refractive index, absorption coefficient, absorption bandgap, and resistivity of the Gd-doped indium oxide films are decreased, which indicates that the Gd-doped indium oxide can be used as a transparent conductive oxide. In the n-p junction based photo-detector devices, the current-voltage curves are highly related Gd concentration. Besides, the Magnetic hysteresis loops are also related to the Gd concentration. My comments and suggestions are listed as follows.

1.     The caption of Figure I is not related to the figure.

2.     The authors are suggested to explain the trend of the RMS values calculated from the AFM images.

3.     In Figure 5, it seems that there are many small figures in the Figure 5.

4.     The authors are suggested to explain the trend of the peak width values in Figure 10. Besides, it is suggested to explain shoulder at about 750 nm.

5.     The authors are suggested to explain the trend of the highest extinction coefficient values in Figure. 11. Besides, it is suggested to explain the ripples in the wavelength from 400 nm to 800 nm.

6.     It is suggested to explain the Gd concentration dependent J-V curves.

Author Response

We would like to thank the reviewers for their valuable comments, which helped to improve our manuscript. Below we present one-to-one replies to the comments raised by the reviewers, and summary of the changes made throughout the manuscript. For the sake of convenience, the changes in the text have been added/edited in the yellow color.

Comment#1: The caption of Figure I is not related to the figure.

Response: We are sorry for this mistake. The caption is corrected as ”Figure I: Scheme of the electron beam evaporation set”

Comment#2: The authors are suggested to explain the trend of the RMS values calculated from the AFM images.

Response: The RMS is calculated as the root mean square of the measured surfaces of the microscopic peaks and valleys, i.e. it represents the root mean square value of ordinate values within the definition area. It is comparable to the height of the standard deviation. The peaks and valleys of the rooftops are measured individually for each value, but the measurements are applied to a separate formula. Examining the calculations reveals that the RMS value is impacted by a single significant peak or fault inside the microscopic surface texture. In reference [22], more information regarding the trend of the RMS values derived by the AFM image is provided.

This comment is added at lines 156-160.

Comment#3: In Figure 5, it seems that there are many small figures in the Figure 5.

Response: In figure 5, it appears that there are many small shapes in Figure 5. They may be attributed to the presence of the doping Gd in addition to the matrix material In2O3.

This comment is added at lines # 155-156.

Comment#4: The authors are suggested to explain the trend of the peak width values in Figure 10. Besides, it is suggested to explain shoulder at about 750 nm.

Response: The following explanation is added between lines 190 and 195:

“The peak width of the refractive index (700 < λ < 800) in figure 10 is determined by the transmission of light via nano-holes and the negative phase shift brought on by surface plasmons (SP) scattering at the interfaces between the nano-hole and the substrate. SPs, which are collective oscillations of metal-free electrons trapped at metal-dielectric interfaces and stimulated by an electromagnetic field that is incident on them, are confined in the metal surface. The multiple optical resonance peaks that appear as a shoulder at 750 nm are caused by the coupling and decoupling process between the SP resonance evanescent waves and the incident light through the nano-hole. Reference [24] provides additional information on SPs and plasmon resonance.

Comment#5:  The authors are suggested to explain the trend of the highest extinction coefficient values in Figure. 11. Besides, it is suggested to explain the ripples in the wavelength from 400 nm to 800 nm.

Response: The following explanation is added between lines 195 and 202:

The refractive index and extinction coefficient for all samples decrease as the wavelength increases, as seen in Figs. 8(a) and (b). Light scattering and the decline in absorbance are the causes of this phenomenon. The refractive index and extinction coefficient in the visible region decrease as the Gd content increases. The extinction coefficient value in Fig. 8(b) is rather high. This demonstrates the substantial dielectric loss of the Gd-doped indium oxide thin films. The polycrystallanity of the films was indicated by ripples (interference patterns) in the extinction coefficient spectrum in the wavelength range of 400 nm to 800 nm. The nano-hole form and nano-hole periodicity allow for exact control of the transmission wavelength location and intensity. For instance, the contribution from the structural margins becomes increasingly substantial in short-range systems with few holes, resulting in unique emission patterns.”

Comment#6:  It is suggested to explain the Gd concentration dependent J-V curves.

Response: Description of figure 13 of the J-V curves are modified at lines 267-274 to include explanation sa follows:

With applied voltage in the recommended ranges, figure 13 shows the dark (J-V) characteristics of the produced diode in forward and reverse bias in (In1-xGdx)2O3 thin films on silicon substrate. It is obvious that up until a level of 8%, the current density rises along with the Gd content before beginning to significantly be fixed at a level of 10%. Figures 13(a) and (b) depict the relationship between the forward and reverse biases of the applied voltage in the dark and in the light, respectively. The forward bias voltage's current is higher than the reverse bias voltage's current (J-V). These figures show how an increase in the forward bias behavior of the solar cell results in an increase in the current density behavior, which significantly increases in the low voltage area. In the depletion zone, also known as the “low voltage region”, the reverse current density of the examined produced p-n junction displays a weaker exponential behavior than the junction's forward current density does in the same region. As a result, it may be argued that the constructed p-n junction has amazing rectification qualities because as the resistivity falls, the Gd concentration rises. The measurements demonstrate the n-type nature of the (In1-xGdx)2O3 films. It is clear that when the Gd content rises, the carriers' concentration, n and mobility change. The mobility is greatest for both carrier concentrations at a Gd concentration of 8 at%, and becomes roughly fixed at 10 at%. These findings show that the resistance diminishes with increasing the carrier concentration. A decrease in the crystalline size and an increase in the lattice strain are related to an increase in mobility. For instance, (In0.92Gd0.08)2O3 films that are candidates for optoelectronic and solar cell applications have fair crystal light size, high conductivity, high carrier concentration, and carrier mobility.

Reviewer 2 Report

The manuscript entitled Structural, Optical, Electric and Magnetic Characteristics of (In1-xGdx)2O3 Films for Optoelectronics by Moustafa et al demonstrated the possibility whether it is feasible to dope Gd into the In2O3 host lattice. In my opinion, this work is of interest to researchers in the field of a promising approach to Gd-doped In2O3 films are suitable candidates for spintronic device production at ambient temperature. 

As just a minor revision, the authors should consider the following comments to improve their manuscript. 

Page 3 Ling 86 Figure I should be modified.

Figure 1 and Figure 2 should be merged and explain it in revised manuscript.

Figure 7 and Figure 8 should be merged and explain it in revised manuscript.

Figure 10, 11, and 12 should be merged and explain it in revised the manuscript.

Author Response

We would like to thank the reviewers for their valuable comments, which helped to improve our manuscript. Below we present one-to-one replies to the comments raised by the reviewers, and summary of the changes made throughout the manuscript. For the sake of convenience, the changes in the text have been added/edited in the yellow color.

Comment#1: Page 3 Ling 86 Figure I should be modified.

Response: The caption is corrected to: “Figure I: Scheme of the electron beam evaporation set.”

Comment#2: Figure 1 and Figure 2 should be merged and explain it in revised manuscript.

Response: We merged Figure 1 and Figure 2 and explained them in revised manuscript.

Please, see the revised version of the manuscript.

Comment#3: Figure 7 and Figure 8 should be merged and explain it in revised manuscript.

The response

Response: We merged Figure 7 and Figure 8 and explained them in revised manuscript.

Please, see the revised version of the manuscript.

Comment#4: Figure 10, 11, and 12 should be merged and explain it in revised the manuscript.

Response: We merged Figure 10, 11, and 12 and explained it in revised manuscript.

Please, see the revised version of the manuscript.

Round 2

Reviewer 1 Report

In the SEM iamges, it seems that there are many small pictures embedded in Figure 4 (a) and (b). Please check it carefully.

Reviewer 2 Report

The authors have improved the manuscript for publication.

Author Response

Deep thanks to the reviewer